# Compact Quad Band MIMO Antenna Design with Enhanced Gain for Wireless Communications

**DOI:** 10.3390/s22197143

**Published:** 2022-09-21

**Authors:** Sanjukta Nej, Anumoy Ghosh, Sarosh Ahmad, Adnan Ghaffar, Mousa Hussein

**Affiliations:** 1Department of Electronics and Communication Engineering, NIT Mizoram, Aizawl 796012, India; 2Department of Signal Theory and Communications, Universidad Carlos III de Madrid (UC3M), 28911 Leganes, Spain; 3Department of Systems and Computer Engineering, Carleton University, Ottawa, ON H4N 1L1, Canada; 4Department of Electrical and Electronic Engineering, Auckland University of Technology, Auckland 1010, New Zealand; 5Department of Electrical Engineering, United Arab Emirates University, Al Ain 15551, United Arab Emirates

**Keywords:** quadband MIMO, meander-line, gain, isolation, CCL

## Abstract

In this paper, a novel microstrip line-fed meander-line-based four-elements quad band Multiple Input and Multiple Output (MIMO) antenna is proposed with a gain enhancement technique. The proposed structure resonates at four bands simultaneously, that is, 1.23, 2.45, 3.5 and 4.9 GHz, which resemble GPS L2, Wi-Fi, Wi-MAX and WLAN wireless application bands, respectively. The unit element is extended to four elements MIMO antenna structure exhibiting isolation of more than 22 dB between the adjacent elements without disturbing the resonant frequencies. In order to enhance the gain, two orthogonal microstrip lines are incorporated between the antenna elements which result in significant gain improvement over all the four resonances. Furthermore, the diversity performance of the MIMO structure is analyzed. The Envelope Co-Relation Coefficient (ECC), Diversity Gain (DG), Channel Capacity Loss (CCL), Mean Effective Gain (MEG) and Multiplexing Efficiency are obtained as 0.003, 10 dB, 0.0025 bps/Hz, −3 dB (almost) and 0.64 (min.), respectively, which are competent and compatible with practical wireless applications. The Total Active Reflection Coefficient (TARC) resembles the characteristic of the individual antenna elements. The layout area of the overall MIMO antenna is 0.33 λ × 0.29 λ, where λ is the free-space wavelength corresponding to the lowest resonance. The advantage of the proposed structure has been assessed by comparing it with previously reported MIMO structures based on number of antenna elements, isolation, gain, CCL and compactness. A prototype of the proposed MIMO structure is fabricated, and the measured results are found to be aligned with the simulated results.

## 1. Introduction

The concept of MIMO antenna has been reviewed and gaining impetus due to its ability to achieve high data rate, low power consumption as well as low latency [1,2]. MIMO antenna is extensively used in 5G mobile phones for higher data rate and low latency. The same goes for other communication networks like machine-to-machine connectivity and large wireless communication frameworks due to high channel capacity as well as high data speed [3,4,5]. Recently, MIMO antenna technology served different applications such as digital television, WLAN, mobile communication services, vehicle-to vehicle communication, etc. [6,7,8]. Since any industry wants to provide multiple wireless services through multiple frequency bands, keeping in mind the requirements of high data rate and low latency, multiband MIMO antennas are in high demand. Moreover, as electronic devices are becoming miniaturized progressively, the space constraint dictates that compact MIMO structures are to be designed so that numerous antenna elements can be accommodated within a given layout area.

In the domain of MIMO antenna design, Dual band two port [9,10] and four port [11] MIMO antennas have been investigated which have high mutual coupling effect with low gain at operational bands. MIMO systems operating above dual band generally have two elements proposed in [12,13,14,15,16]. A four elements triple band MIMO consisting of microstrip line fed square-ring slot radiator antenna has been designed which has 17 dB isolation with a large layout area which becomes disadvantageous for practical uses [17]. A four elements MIMO with 8 ports configuration has been designed which operates three microwave and two mm wave range applications, offers an average isolation of 16 dB and have an ECC value of 0.16 [18]. A MIMO antenna with a high density of up to eight elements is proposed in [19]. As the number of elements increases, the mutual coupling effect becomes prominent. Various techniques such as integrating stub at ground and at feed line [12], etched open ended slot at the ground plane [14], multiple techniques like parasitic elements and defected ground structure [13] and U-shaped slot between adjacent antenna pairs [20] have been incorporated to improve the inter antenna element isolation for multiband MIMO systems. Gain enhancement techniques like employing DGS [21,22] and metamaterial superstrate [23] are used for single band antennas. An artificial magnetic conductor underneath the antenna structure is used to enhance the gain of a dual band MIMO structure [20].

Multiband MIMO antennas for mobile/wireless communication are more desirable than wide band antennas to mitigate the unnecessary interference from unused bands. Previous literature on multiband MIMO antennas with more than two elements primarily encompasses two or three wireless bands [10,11,12,13]. Furthermore, gain enhancement of multiband MIMO antennas is a challenge which has been countered by the use of Artificial Magnetic Conductors (AMC) [20]. The use of AMC as a separate layer with an air gap increases the profile of the antenna thus making it unsuitable for very sophisticated systems.

The authors have addressed these problems in the proposed work wherein a quad-band four elements MIMO antenna has been proposed. The use of four elements is more advantageous than that of two elements in terms of Channel Capacity. Moreover, the use of orthogonal microstrip lines on the same plane as the MIMO radiators instead of using any separate frequency selective structure as a separate layer for gain enhancement has given the proposed MIMO antenna a very low profile which can be easily integrated with sophisticated systems. The surface current density at the resonant frequencies is investigated to understand the generation of quad-resonance and the gain enhancement mechanism. The diversity performance of the antenna is detailed with respect to the TARC, ECC, DG, CCL, MEG and Multiplexing efficiency characteristics to understand the MIMO behavior. All the simulation results pertaining to the proposed design are obtained by using the full-wave simulation software ANSYS HFSS v. 20. The simulation results are experimentally validated through fabrication and measurement of the prototype of the proposed structure.

## 2. Antenna Design

### 2.1. Meander-Line Antenna Element Design

The evolution of single element meander-line antenna design has been given in Figure 1a–c. The|S11| characteristics according to the design evolution have been presented in Figure 1d. From Figure 1d, it is inferred that adding a meander-line microstrip structure on the left-hand side (Antenna 1) gives a single resonance at 2.45 GHz. Addition of another meander line on the right-hand side (Antenna 2) gives two extra resonances, one at 3.5 GHz and another wideband response encompassing 4.2 GHz–5.1 GHz. A small microstrip line is appended below the right-hand meander line (Antenna 3) to get a fourth resonance at 1.23 GHz along with bandwidth adjustment at 4.9 GHz.

The final antenna design comprises multiple unequal length microstrip arms which are arranged in meander-line form. The microstrip feedline has been presented in Figure 2. FR4 substrate with dielectric constant 4.4, loss tangent 0.02 and height 1.6 mm is used. The relevant dimensions of the single element meander-line antenna are given in Table 1. The meander-line structure provides longer current path than a conventional microstrip line in a given layout area thereby achieving compactness.

For any given meander line length L_phy_, the corresponding resonant frequency f is given by
(1)$$f=\frac{c}{2L_{\mathrm{total}}\sqrt{\varepsilon_{\mathrm{eff}}}}$$

Here, the total length of each meander line section is given as
(2)$$L_{\mathrm{total}}=L_{\mathrm{phy}}+\Delta L$$

Here, c is the electromagnetic wave velocity in free space, and ΔL is the extra length due to fringing fields at the meander line edges given by [24]
(3)$$\Delta L=t\left(\frac{0.412\left(\varepsilon_{\mathrm{eff}}+0.3\right)\left(\frac{w_{m}}{t}+0.264\right)}{(\varepsilon_{\mathrm{eff}}-0.258)\left(\frac{w_{m}}{t}+0.8\right)}\right)$$
where $\varepsilon_{\mathrm{eff}}$ represents the effective dielectric constant due to quasi-TEM mode of microstrip line given by [25]
(4)$$\varepsilon_{\mathrm{eff}}=\left(\frac{\varepsilon_{r}+1}{2}\right)+\left(\frac{\varepsilon_{r}-1}{2}\right)\left(\frac{1}{\sqrt{1+\frac{12t}{w_{m}}}}\right)$$

Here, $\varepsilon_{r}$ and t are the relative permittivity and thickness of the dielectric substrate, respectively, and $w_{m}$ is the width of the meander-line strips of the proposed antenna structure. The structure has four distinct meander-line current patches to generate four resonant frequencies at 1.23, 2.45, 3.5 and 4.9 GHz. The simulated current distribution patterns at the resonant frequencies are shown in Figure 3. It is observed that for each resonance, a particular meander-line section is accommodating some current minima (node) and maxima (anti-node). The distance between two successive nodes is $\lambda_{g}/2$, where $\lambda_{g}$ is the guided wavelength given by [24]
(5)$$\lambda_{g}=c/\left(f\sqrt{\varepsilon_{\mathrm{eff}}}\right)$$

Equation (5) gives the resonant frequency by observing the location of current nodes and anti-nodes at any particular section of a meander-line responsible for that particular resonance. For the first resonance, the distance between two successive nodes is obtained from Figure 3a with reference to Figure 2 for dimensional parameters as
(6)$$L_{1}=P_{1}+2P_{2}+P_{3}+P_{4}+S_{1}+S_{2}+S_{3}+S_{7}-5w_{m}$$

Similarly, for the second, third and fourth resonances, the distance between two successive nodes is obtained as L_2_, L_3_, L_4_, respectively, from Figure 3b–d subsequently given by
(7)$$L_{2}=P_{5}+P_{6}+S_{4}+S_{6}-2w_{m}\;$$
(8)$$L_{3}=P_{3}+P_{4}+0.2S_{2}+S_{3}\;$$
(9)$$L_{4}=P_{1}+S_{1}-w_{m}$$

### 2.2. Quad Elements MIMO Antenna Design

The single antenna is extended to quad-element MIMO structure as illustrated in Figure 4a. The orientation of the antenna elements is adjusted to have good isolation between the antenna elements without significant inter-element distance. In order to enhance the gain of the structure, a cross parasitic microstrip structure is inserted between the antenna elements as depicted in Figure 4b. The pertinent dimensions related to Figure 4 are given in Table 2.

## 3. Results and Discussions

### 3.1. Single Antenna Performance

The parametric adjustment has been observed by varying the length of the meander-line section. Further simulations have confirmed that P_1_ and P_5_ parameters (Referring to Figure 2) have most significant effects on the |S11| response of the antenna. The results are illustrated in Figure 5 and indicate that as P_1_ is increased, all the resonant frequencies shift towards the left except the second resonance. As the parameter P_5_ is increased, only the second resonance shifts towards the left.

After final adjustment of the length of the meander-line sections, the|S11| plot of the single element antenna is presented in Figure 6, which demonstrates that the structure resonates at 1.23 GHz (1.21–1.24 GHz), 2.47 GHz (2.42–2.5 GHz), 3.5 GHz (3.4–3.6 GHz) and 4.95 GHz (4.8–5.1 GHz) with good impedance matching and percentage bandwidths of 2.43, 3.23, 5.71 and 6.06, respectively. Table 3 represents a comparison study of resonant frequencies between simulated response and theoretical calculation mentioned in subheading 2.1 which shows close resemblance between the values obtained from simulation and current distribution analysis.

### 3.2. MIMO Antenna Performance

The antenna performances in terms of scattering parameters and gain have been obtained by simulating the MIMO antenna with excitation at one port while other ports are connected to matched load. While simulating the |Sii| response, the ith port is excited, and all other ports are matched. Figure 7a highlights that the |Sii| characteristic is almost similar for all the individual antenna elements and resembles the single element response. The value of |Sii| is below −20 dB for all three resonant frequencies and −15 dB for the fourth resonant frequency, which shows that a good impedance matching has been achieved over all the ports. Moreover, the addition of parasitic microstrip lines does not perturb the |Sii| response. In order to characterize the isolation property of the antenna, |Sij| has been simulated with excitation ports assigned to ports i and j while other ports are matched. From Figure 7b, it is observed that the isolation between any two adjacent elements is more than 22 dB at all the resonant bands for the MIMO structure with and without parasitic elements. A very low mutual coupling has been accomplished without involving any additional structure and within a compact layout of 0.37 λ × 0.32 λ, where λ is the free-space wavelength corresponding to the lowest resonance.

Referring to Figure 4b, increasing the parameter Sm increases the gain of the first resonance slightly whereas the gain at other resonances deteriorates. As the parameter St is increased, the gains at the second and third resonances are enhanced but the gain falls drastically at the first and fourth resonances. Hence, the optimum length is as proposed in the manuscript. Table 4 presents a comparative study of the MIMO antenna with and without the parasitic microstrip elements with respect to gain. It reveals that the parasitic elements increase the gain of the antenna by 1.06, 1.42, 2.48 and 3.09 dB at 1st, 2nd, 3rd and 4th resonances, respectively.

The simulated surface current density diagrams of the MIMO structure with parasitic microstrip lines are presented in Figure 8 to describe the mechanism of gain enhancement which takes into account both the direction and magnitude of the surface current density vectors. The figures represent the vector summation of the current densities in the meander line antenna and parasitic microstrip lines at all four resonances while the port 1 is excited. Here, J_X_Antenna_, J_Y_Antenna_ denote the resultant surface current density vector components at the X direction and Y direction, respectively, for the microstrip patch antenna. Similarly, J_X_Strip_ and J_Y_Strip_ denote the resultant surface current density vector components at the X direction and Y direction for the parasitic microstrip section, respectively. From Figure 8a, it is observed that the direction of vector summation of horizontal (X-axis) and vertical (Y-axis) current flow in the parasitic crossed microstrip lines and meander-line segment responsible for the first resonance are in the same direction. Hence, the radiation due to the current induced from the parasitic microstrip line complements the radiation of the antenna, thereby enhancing the gain. Due to the same reason, for the other three resonances as shown in Figure 8b–d, the gain is enhanced substantially. It has been observed in Table 4 that the magnitude of the improvement in gain varies for the different resonances as the magnitudes of the resultant current vectors in the parasitic microstrips vary with the resonances. Therefore, the magnitude of constructive interference of the radiated waves from the antenna and the parasitic microstrips in the far field differ at the resonant frequencies with strongest constructive interference resulting in the highest enhancement in gain.

### 3.3. Diversity Performance

The diversity performances of the MIMO structure are described by Envelop Correlation Coefficient (ECC), Diversity Gain (DG), Channel Capacity Loss (CCL), Total Active Reflection Coefficient (TARC), Mean Effective Gain (MEG), Multiplexing Efficiency. ECC is the measure of the effect of one antenna element on another due to coupling which is given by [2]
(10)$$\rho_{12}=\frac{{\left|S_{11}S_{12}+S_{22}S_{21}+S_{13}S_{32}+S_{14}S_{42}\right|}^{2}}{\left(1-\left|S_{11}{}^{2}+S_{21}{}^{2}+S_{31}{}^{2}+S_{41}{}^{2}\right|\right)\left(1-\left|S_{21}{}^{2}+S_{22}{}^{2}+S_{32}{}^{2}+S_{42}{}^{2}\right|\right)}$$

Here, $\rho_{12}$ is the description of the isolation or correlation between the antenna element pairs Ant 1, Ant 2. Similarly, $\rho_{13}$, $\rho_{14}$ are calculated for the antenna element pairs Ant 1, Ant 3 and Ant 1, Ant 4, respectively, and plotted in Figure 9a. The ECC value is <0.003 over the operational bandwidths, as shown in Figure 9a which is much below the maximum acceptable range of 0.5 [9]. Diversity gain can be used to describe the gain enhancement of a multiple-antenna system in a combined signal over time-averaged SNR. DG is obtained from the ECC value as [2]
(11)$$DG=10\sqrt{1-{\left|\mathrm{ECC}\right|}^{2}}$$

The DG response is graphically illustrated in Figure 9b, and it is observed that the value is nearly 10 dB at the resonant frequencies. CCL and TARC (Γ) for quad elements MIMO are obtained from Equations (12) and (13), respectively, as [1,20],
(12)$$\mathrm{CCL}=-\log_{2}\det\left(S\right)$$
where $S=\left(\begin{array}{cc} \sigma_{11} & \sigma_{12}\\ \sigma_{21} & \sigma_{22} \end{array}\right)$,

$\sigma_{\mathrm{ii}}$ = $1-(|S_{\mathrm{ii}}{|}^{2}+{\left|S_{\mathrm{ij}}\right|}^{2}$),

$\sigma_{\mathrm{ij}}$ = $S_{\mathrm{ii}}{}^{*}S_{\mathrm{ij}}+S_{\mathrm{ii}}S_{\mathrm{jj}}{}^{*}$)
(13)$$\mathrm{TARC}=\sqrt{\frac{{\left|S_{11}+S_{12}+S_{13}+S_{14}\right|}^{2}+{\left|S_{21}+S_{22}+S_{23}+S_{24}\right|}^{2}+\left|S_{31}+S_{32}+S_{33}+S_{34}\right|+{\left|S_{41}+S_{42}+S_{43}+S_{44}\right|}^{2}}{4}}$$

For N x N MIMO system, the channel capacity is proportional to N, where the individual sub-channel is not correlated but an uncorrelated channel is not practically possible and hence channel capacity loss is determined [20]. CCL response and TARC are illustrated in Figure 9c,d, respectively. Figure 9c highlights that the CCL is 0.0025 bps/Hz for all the antenna pairs, thus signifying that it is capable of a high data transmission rate which is one of the main advantageous features of MIMO antennas over any standalone antenna. Figure 9d demonstrates that TARC response is almost similar to individual |Sii| response which indicates the overall return loss of the MIMO structure is in acceptable range.

The mean effective gain has been measured for individual ports of quad elements MIMO from the equation given by [13]
(14)$${\mathrm{MEG}}_{i}=0.5\;(1-\sum_{j=1}^{M}{\left|S_{\mathrm{ij}}\right|}^{2})\;$$

The MEG is denoted as the electromagnetic power measurement at individual port of the MIMO in multipath scenario and the value should be −3 ≤ MEG (dB) < −12 [26]. The computed MEG is presented in Figure 9e which shows that the value of the MEG is near −3 dB at all the ports (port 1 to port 4) at the operation band which is advantageous for real time multipath fading scenario.

The multiplexing efficiencies are an important parameter for a multiport or array antenna structure which simultaneously encounters the total efficiency of two ports and correlation between them. The multiplexing efficiency is given by [22]
(15)$${\eta \mathrm{MUX}}_{\mathrm{ij}}=\sqrt{\eta_{i}\eta_{j}\left(1-\rho_{\mathrm{ij}}\right)}$$

Here, $\eta_{i}{\;\mathrm{and}\;\eta }_{j}$ are the total efficiency at ith and jth ports, and $\rho_{\mathrm{ij}}$ is the envelop correlation coefficient between the ith and jth ports. The highest multiplexing efficiency value obtained is 0.82 at the fourth resonance as shown in Figure 9f. The overall value is between 0.64 to 0.73 at other operational bands as represented in Figure 9f.

### 3.4. Measurement Results

The prototype of the final design of the quad elements quad band MIMO with enhanced gain is fabricated using a PCB prototype machine MITS Eleven Lab on a FR4 substrate. The photograph of the fabricated structure is shown in Figure 10. The |Sii| and |Sij| characteristics of the fabricated prototype are measured using a Vector Network Analyzer of make Keysight Technologies N5235A and compared with the simulated result and presented in Figure 11a,b, respectively. From the figure, it is seen that the measured results almost resemble the simulated results except at the fourth resonance where some minor discrepancies are observed which may be attributed to fabrication error. Since the measured |Sii| and |Sij| characteristics are almost similar to the simulated results, the diversity performance of the fabricated MIMO antenna will also resemble the simulated results since diversity performance is derived from the |Sii| and |Sij| values.

The measured resonating frequencies cover the targeted wireless operational bands as GPS L2, Wi-Fi, Wi-MAX and WLAN and closely resemble the simulated results as shown in Table 5. The table also compares the simulated gain with the measured gain to validate the simulated results. Gain is measured using the technique described in [27]. Since gain is measured in free space without an Anechoic Chamber, the minor mismatch with simulated results is due to multipath reflections and loss from adapters. The simulated and measured normalized radiation patterns for all four resonances at both the XZ plane and YZ plane are exhibited in Figure 12. The radiation patterns are measured for port 1 where other ports are connected to the 50 ohm load. Only port one is considered for measuring the antenna characteristics as the four ports are placed in similar way at X and as well as Y direction. The figure illustrates that both the simulated and measured patterns are in good accord. The half power beamwidths (HPBW) at the XZ plane are 110°, 76°, 72° and 96° for the 1st, 2nd, 3rd and 4th resonances, respectively, and at the YZ plane, they are 118°, 97°, 112°, 111° for the 1st, 2nd, 3rd and 4th resonances, respectively. Most of the radiation patterns are at the boresight direction with high half power beam width except in the third and fourth frequencies at the XZ plane where the major lobe is tilted as shown in Figure 12c,d, respectively. Although crosspolar radiation increases with frequency, it is maintained below −15 dB in the boresight direction.

A comparison table of different characteristics of the proposed work with existing MIMO antenna designs is given in Table 6 to highlight the advantage of the proposed structure. In terms of compactness, the proposed structure is most compact except [12,14] although in [12,14] the number of antenna elements is confined to two. The minimum isolation of [22] is better than the proposed structure, but in [22] the layout area is significantly more than the proposed antenna as a result of which the inter-element distance is higher which gives better isolation. The maximum gain of the proposed structure is substantially higher than all other compared structures due to its efficient gain enhancement technique. CCL is an important parameter which signifies the data handling capacity of a MIMO antenna. Thus, this parameter is required to be as low as possible and in this perspective, the proposed structure has the best performance except compared to [22] although [22] is a single frequency MIMO antenna. In terms of multiband performance, the proposed structure gives four resonances which is less than only [16,18] although the proposed structure is superior to [16,18] with respect to other compared parameters. The peak gain obtained in [18] is high although the high gain has been achieved at the very high mmwave frequency of 28 GHz. Thus, considering the multiband operability sub 6 GHz band with more than two antenna elements along with high isolation between antenna elements without compromising the compactness, exhibiting high gain and data handling capability, the proposed structure is the best candidate for MIMO applications as compared to previously reported multiband MIMO antennas.

## 4. Conclusions

A highly compact quad elements quad band MIMO antenna with enhanced gain by implementing parasitic microstrip line has been presented. The asymmetric meander-line patch forms the basic antenna element of the MIMO structure which provides four independent current paths for four resonating frequencies, 1.23, 2.45, 3.5 and 4.9 GHz, which resemble the GPS L2 band, Wi-Fi, Wi-MAX band and WLAN wireless application bands. The MIMO structure has been configured by the orientation of four antenna elements in a side-by-side and face-to-face arrangement. The minimum isolation achieved without any additional structure is greater than 22 dB with minimum inter element spacing of 0.028 λ. A gain enhancement technique has been introduced where a parasitic orthogonally crossed microstrip line is incorporated within the MIMO layout that substantially enhances the gain at all the four resonant frequencies without disturbing the S-parameters of the antenna. The proposed gain enhanced MIMO antenna has been fabricated, and the measured results are in compliance with the simulation results. The ECC value is below 0.003 over the entire frequency range, and the CCL value is near 0.0025 bps/Hz at resonant frequencies, which indicates that the diversity responses are good for practical MIMO applications. The MEG and Multiplexing efficiency are computed for all ports which become advantageous over the operational bandwidth for the proposed MIMO antenna. The TARC plot is similar to the |Sii| graph which demonstrates good impedance matching in the overall MIMO system. Thus, the proposed work gives a compact design of a high gain multiband MIMO antenna with high data handling capability, thereby having the potential to be used in practical wireless MIMO systems.

## Figures and Tables

**Figure 1 sensors-22-07143-f001:**
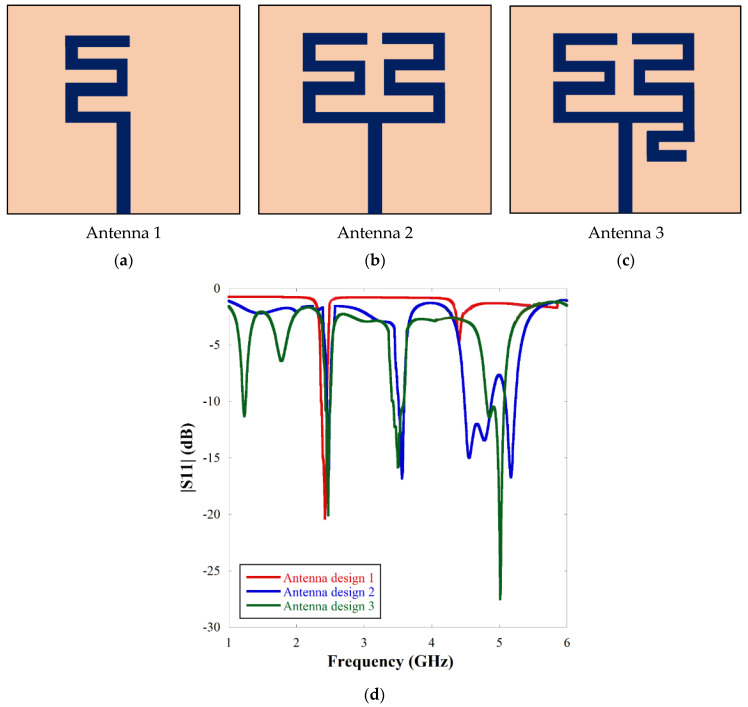
The evolution of meander-line antenna: (**a**) first step, (**b**) second step, (**c**) third step of antenna design and (**d**) |S11| characteristics according to the design evolution.

**Figure 2 sensors-22-07143-f002:**
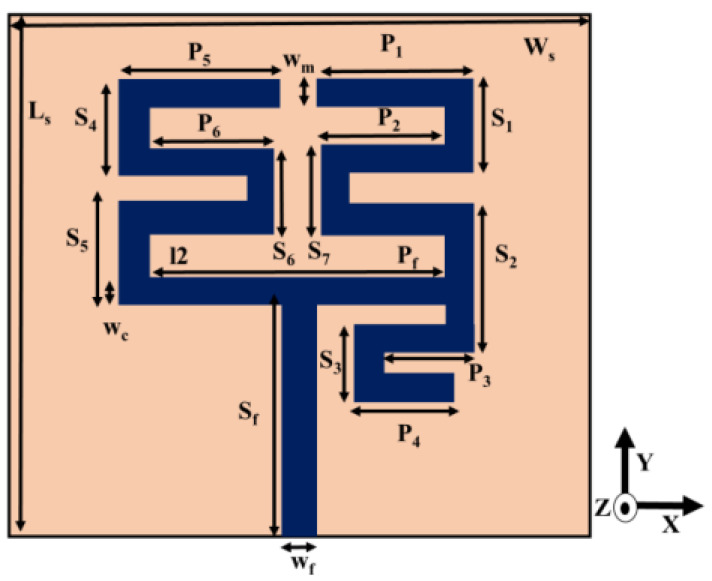
Schematic diagram of single unit quad band meander-line antenna.

**Figure 3 sensors-22-07143-f003:**
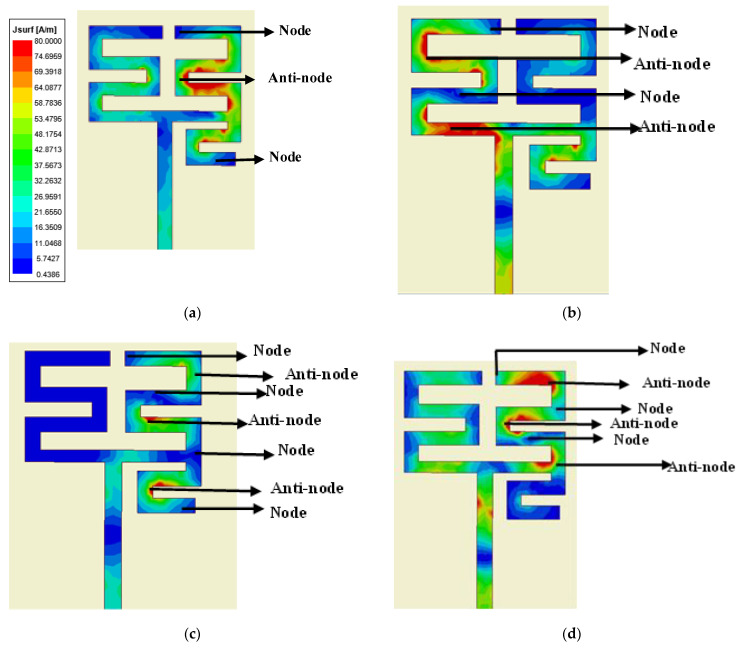
Simulated current distribution at (**a**) 1st resonance, (**b**) 2nd resonance, (**c**) 3rd resonance, (**d**) 4th resonance of single unit quad band meander-line antenna.

**Figure 4 sensors-22-07143-f004:**
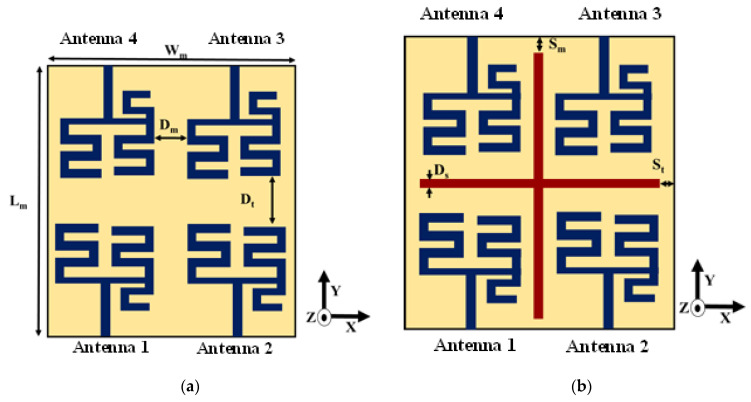
Schematic diagram of four element MIMO antenna (**a**) without parasitic cross microstrip line and (**b**) with parasitic cross microstrip line.

**Figure 5 sensors-22-07143-f005:**
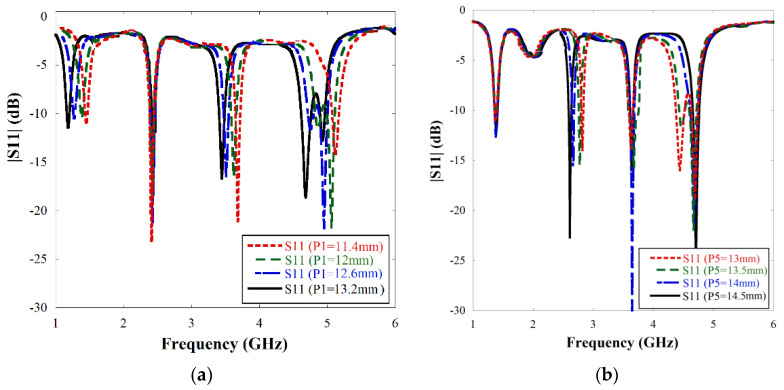
|S11| graph of the single unit meander line antenna variation of length of (**a**) P_1_ and (**b**) P_5_.

**Figure 6 sensors-22-07143-f006:**
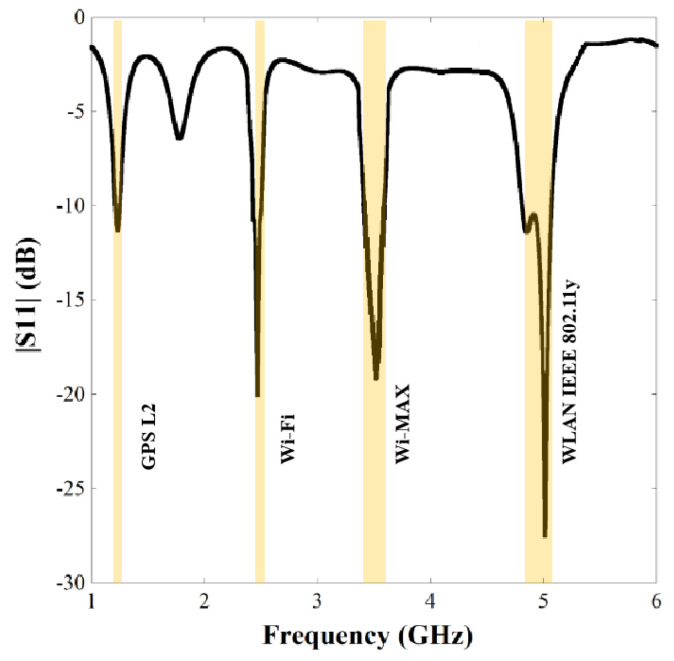
|S11| graph of the single unit meander line antenna.

**Figure 7 sensors-22-07143-f007:**
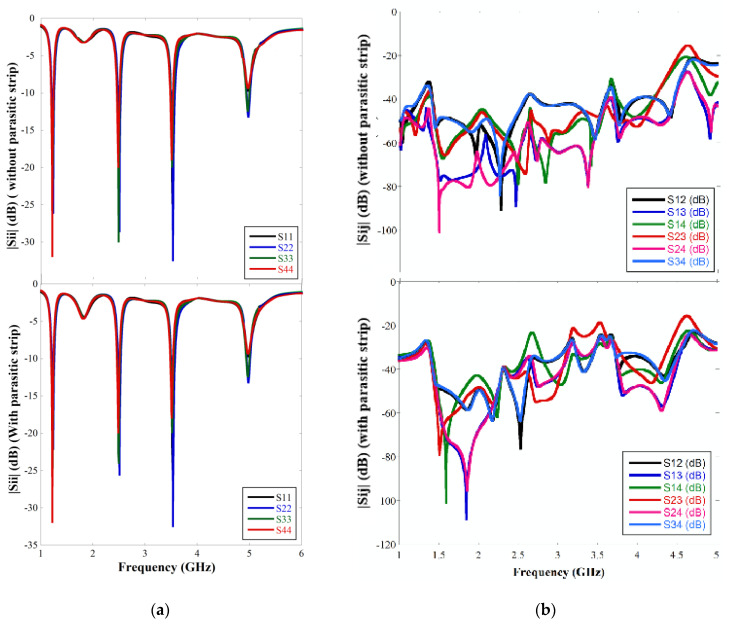
(**a**) |Sii| graph and (**b**) |Sij| graph of the four element MIMO with and without parasitic cross microstrip line.

**Figure 8 sensors-22-07143-f008:**
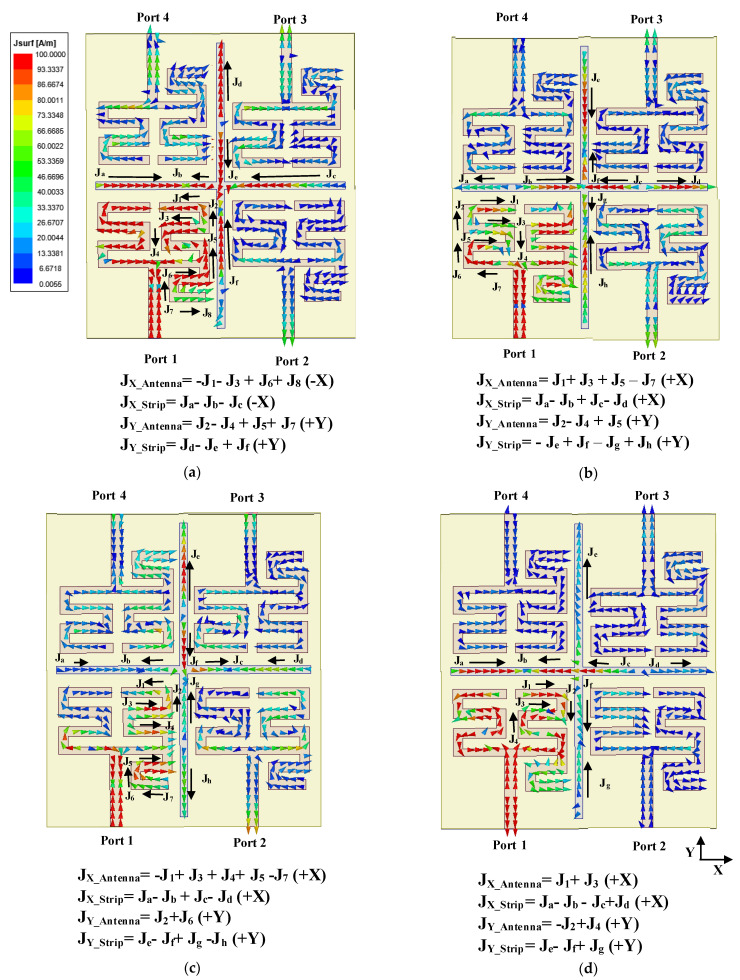
Current distribution of MIMO antenna with cross parasitic microstrip line structure at (**a**) 1st resonance, (**b**) 2nd resonance, (**c**) 3rd resonance and (**d**) 4th resonance.

**Figure 9 sensors-22-07143-f009:**
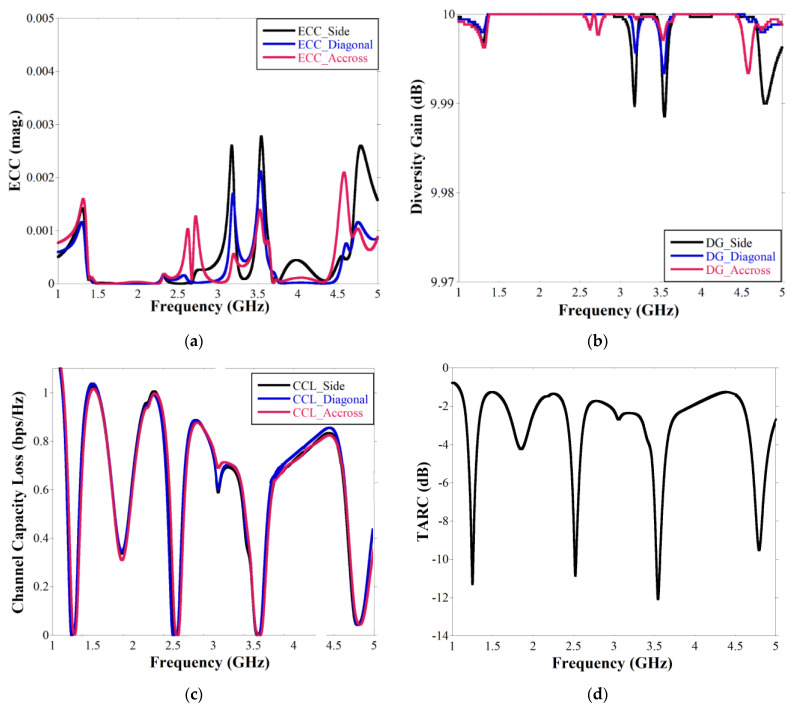
Diversity performances of quad elements quadband MIMO antenna (**a**) ECC, (**b**) diversity gain, (**c**) CCL, (**d**) TARC, (**e**) MEG, (**f**) multiplexing efficiency plot.

**Figure 10 sensors-22-07143-f010:**
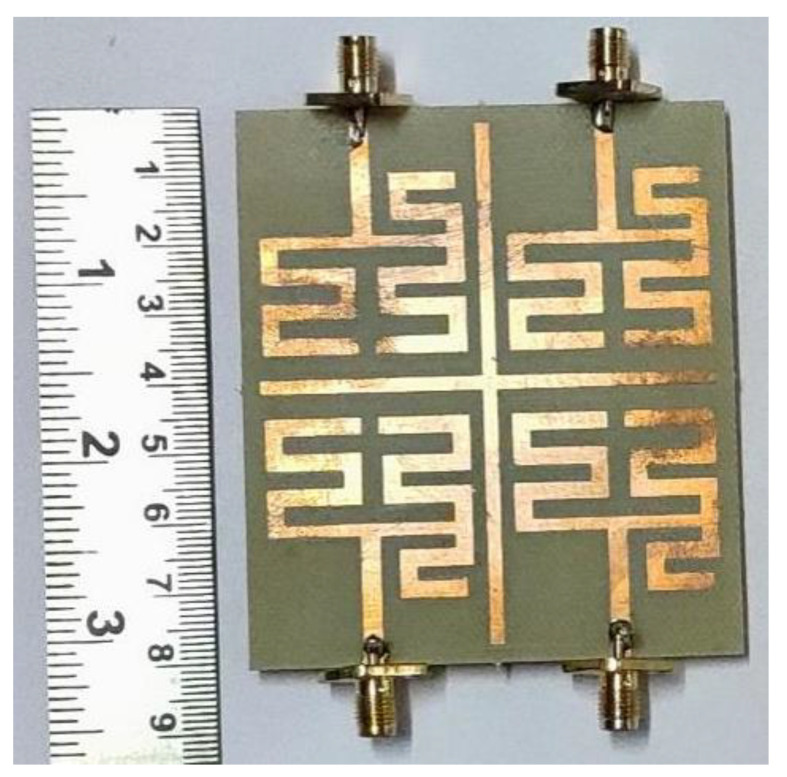
Image of fabricated four elements MIMO with parasitic cross microstrip line structure.

**Figure 11 sensors-22-07143-f011:**
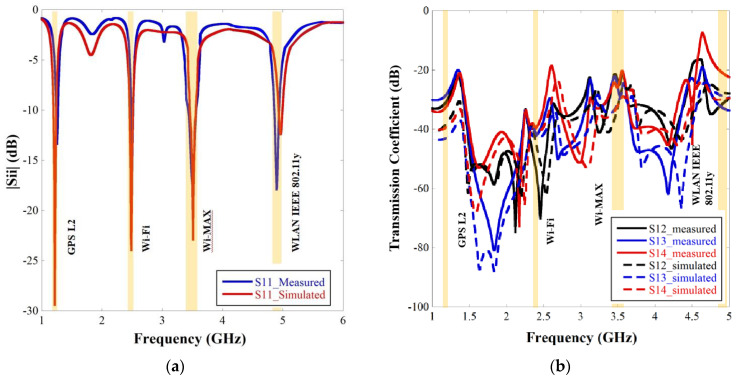
Comparison scattering parameter graphs of the measurement results of the fabricated prototype and simulated results of the four element MIMO with parasitic microstrip line: (**a**) |Sii| graph and (**b**) Transmission coefficient graph.

**Figure 12 sensors-22-07143-f012:**
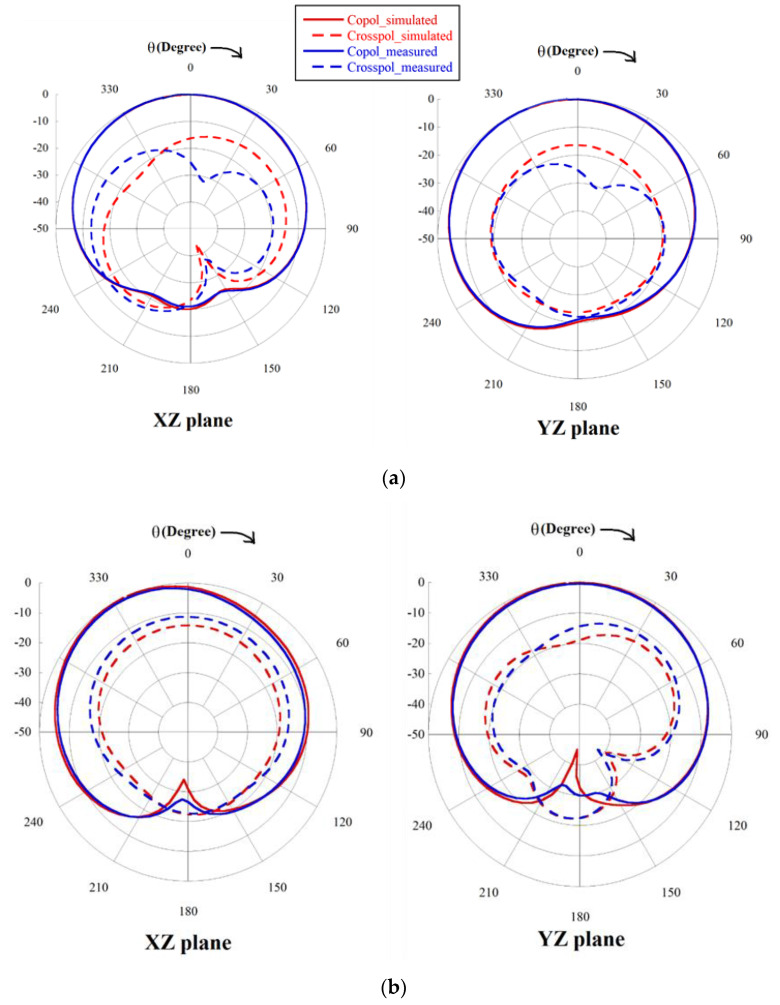
Normalized copolar and crosspolar radiation pattern of the four element MIMO at (**a**) 1st resonance, (**b**) 2nd resonance, (**c**) 3rd resonance and (**d**) 4th resonance.

**Table 1 sensors-22-07143-t001:** Design parameters of single unit meander-line antenna.

Parameter	L_s_ × W_s_	w_m_	w_c_	w_f_	S_1_	S_2_	S_3_	S_4_	S_5_	S_6_	S_7_	S_f_	P_1_	P_2_	P_3_	P_4_	P_5_	P_6_	P_f_
Dimension (mm)	60 × 60	2.5	2.0	2.8	9	11.5	7	8.5	7.5	7.5	7	25.5	12.6	10	8	7	14	11	24

**Table 2 sensors-22-07143-t002:** Design parameters of four elements MIMO antenna.

Parameter	Dimension (mm)
L_m_ × W_m_	80 × 70
D_m_	6
D_t_	9
S_m_	2.5
S_t_	2.5
D_s_	2

**Table 3 sensors-22-07143-t003:** Comparison chart between calculated and simulated resonant frequencies of single unit meander-line antenna.

Length of Two Successive Nodes (mm)	Resonant Frequency (GHz)	
	Theoretical	Simulation
67.9	1.227	1.23
34.0	2.45	2.47
23.8	3.5	3.5
17.0	4.9	4.95

**Table 4 sensors-22-07143-t004:** Comparison table of gain parameter of the four element MIMO antenna with and without parasitic cross microstrip line.

Frequency (GHz)	Gain (dB)	
	Without Parasitic Cross Microstrip Line	With Parasitic Cross Microstrip Line
1.23	0.67	1.73
2.45	3.41	4.83
3.5	5.14	7.62
4.95	6.72	9.81

**Table 5 sensors-22-07143-t005:** Comparison table of measured and simulated antenna parameters of the four element MIMO antenna.

Frequency (GHz)	Simulated			Measured			Intended Wireless Application Bands
	Frequency (GHz)	Bandwidth	Gain (dB)	Frequency (GHz)	Bandwidth	Gain (dB)	
First resonance	1.23	40 MHz (1.21–1.24 GHz)	1.73	1.237	30 MHz (1.22–1.25 GHz)	1.67	GPS L2 (1.22–1.23 GHz)
Second resonance	2.45	80 MHz (2.4–2.48 GHz)	4.83	2.42	80 MHz (2.4–2.48 GHz)	4.1	Wi-Fi (2.4–2.48 GHz)
Third resonance	3.5	210 MHz (3.39–3.6 GHz)	7.62	3.48	230 MHz (3.37–3.6 GHz)	7.21	Wi-MAX (3.4–3.6 GHz)
Fourth resonance	4.95	100 MHz (4.9–5.0 GHz)	9.81	4.9	100 MHz (4.85–4.95 GHz)	9.67	WLAN IEEE 802.11y (4.94–4.99 GHz)

**Table 6 sensors-22-07143-t006:** Comparison table of the proposed work with recent work include in references.

Reference	Frequency Band (GHz)	No. of Elements	Total Layout (λ × λ)	Minimum Isolation (dB)	Maximum Gain (dB)	CCL (bps/Hz)
[11]	5.1, 5.7	4	0.42 λ × 0.41 λ	15	2.96	N/A
[12]	2.4, 3.5, 5.25	2	0.26 λ × 0.25 λ	20	4	N/A
[13]	0.9, 1.8, 2.3	2	NA	15.3	4.1	N/A
[14]	2.4, 3.5, 5.5	2	0.2 λ× 0.36 λ	15	3.2	0.4
[16]	4.75, 5.89, 6.74, 8.25, 9.82	2	1.01 λ × 0.55 λ	21.5	N/A	N/A
[17]	2.6, 3.5, 5. 2	4	0.65 λ × 1.29 λ	17	N/A	N/A
[18]	2.25, 2.6, 5.2, 24, 28	4	0.85 λ × 0.85 λ	16	11	N/A
[19]	3.5, 4.8, 5.5	8	NA	10.5	4.8	NA
[20]	3.5, 5.2	2	0.39 λ × 0.39 λ	19	4.7	0.3
[21]	5.3	4	2.56 λ × 1.55 λ	20	9.5	N/A
[22]	5.8	4	1.1 λ × 1.1 λ	30	5.3	0.0023
[23]	5.7	4	1.33 λ × 1.14 λ	20	9.49	N/A
Proposed work	1.23, 2.45, 3.5, 4.9	4	0.33 λ × 0.29 λ	22	9.67	0.0025

λ is the free-space wavelength corresponding to the lowest resonance.

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
