# Peer review of "Compact Quad Band MIMO Antenna Design with Enhanced Gain for Wireless Communications"

_sensors, 2022, doi:10.3390/s22197143_

Round 1

Reviewer 1 Report

This paper proposes a highly compact quad elements quad-band MIMO antenna with increased gain through orthogonal parasitic microstrip lines. However, the article does not explain in detail the optimization and design process of the antenna and microstrip line dimensions.

  1. The article does not mention how to propose the configuration of the antenna, and what are the characteristics and effects of such placement. The design evolution should be provided.
  2. Compared with antennas placed in the same type, what are the advantages and innovations of this antenna?
  3. Can you further explain how the size of the orthogonal parasitic microstrip line affects the antenna gain?
  4. How are the nodes and anti-nodes defined in the article, and what is the relationship between the strength of the current?
  5. In Fig. 9(b), the isolation degree of the experimental results and the simulation results are not consistent, and the specific reason is not given in the paper.
  6. The data curve in the article should indicate the required operating frequency range for easy comparison and analysis.
  7. The differences in the radiation patterns in Fig. 10(a) are not discussed and explained.
  8. Whether the Equation (13) in the article is consistent with that in reference [18].

Author Response

The authors would like to thank all the reviewers for carefully revising the paper and giving constructive suggestions to improve the quality of the paper.  Our response to reviewer comments is attached.

Reviewer 2 Report

1.       The manuscript presents the implementation, experiments, and performance results of a novel microstrip MIMO antenna. In addition, the authors provide a table for comparing the proposal with some state-of-the-art proposals in the open literature.

2.       In what follows, there are some recommendations for the authors:

a.       Please cite some interesting references from this Journal, for example, https://www.ncbi.nlm.nih.gov/pmc/articles/PMC7284607/

b.       It is better to use the term “meander-line” antenna instead of "meanderline" o "meander line" antenna.

c.       The introduction (for example in line 73 of your manuscript) can be enriched, and subsequently more attractive if you give some examples of MIMO antenna in 5G communications and how the iPhone covers the requirements of this high-demand technology.

d.       Use the same letter font in the text in order to match the equations (for example the letter “f” in line 103 and equation (1), “c” in line 111 and equation (1) ).

e.       Use phrases like: where E_ff represents …. Instead of “In Eqn. (3), E_ff,…”. Also, E_ff is used in (1).

f.        Verify the structure of Table 4

Author Response

The authors would like to thank all the reviewers for carefully revising the paper and giving constructive suggestions to improve the quality of the paper. Our response to the reviewer is attached.

Reviewer 3 Report

I have carefully reviewed the Review paper entitled “Compact Quad Band MIMO Antenna Design with Enhanced Gain for Wireless Communications”. The work presented in this paper is interesting and timely. Moreover “Quad-elements Quad-band MIMO antenna with some parasitic microstrip lines” makes this work more practical. To be honest, the expression, readability, and analysis of this manuscript still needs to be revised, but in any case, I think it deserves a further major revision before accepting it for publication. I explain some of my reservations in detail below.

Q1: The research motivation and technical challenge of this work are not clear. The authors should explain the main contributions (State of the art) of this research work more clearly.

Q2: What are the main characteristics of quad elements quad band MIMO antenna, their physical significance, and how are they considered in the proposed study?

Q3: The logicality of the introduction part is relatively poor. The paragraph is too long which causes the structure is not clear enough. The following contents should be more explicit: the backgrounds, differences with the existing works, and the research motivations. Furthermore, we recommend the authors to cite more recent papers related to the MIMO and antenna designing, such as " Channel Propagation Characteristics for Massive Multiple-Input/Multiple-Output Systems in a Tunnel Environment [Measurements Corner]", "A compact four-band high-isolation quad-port MIMO antenna for 5G and WLAN applications ", "Statistical Characteristics of 3D MIMO Channel Model for Vehicle-to-Vehicle Communications". 

Q4: The analysis of Fig. 10  lacks corresponding explanation. Moreover, the details and qualitative analysis  is required for co-polar and cross-polar radiation patterns.

Q5: The  parametric adjustment of simulation environment needs to be mentioned clearly.

Q6: There are many typo errors in manuscript, few of them I am mentioning here.

1.     Table 1, page 3, line 101/102, L_s  X W_s / 60 X 60 use multiplication sign, same as used in Abstract for antenna layout 0.33 λ x 0.29 λ

2.     Table 2, page 5, Parameters and dimension’s size

3.     Table 3, page 15, similar recommendation for “Total Layout” .

4.     Table 3, it is suggested to write NA as N/A .

5.     The description J_X_Antenna = —j_1………, etc, in Fig. 6 (a), (b), (c), (d) seems not clear. It is suggested to write in a better quality image…

These are some of the mistakes which I just mentioned; it is suggested to revise the whole paper consciously and remove such errors.

Q7: The technicality and English writing of this manuscript should be improved substantially.

Author Response

(The authors gave the same response as above.)

Round 2

Reviewer 1 Report

The paper has been revised. The quality of the paper can be further improved by incorporating following suggestions:

1. To validate the node and anti-node method, please calculate L1,L2,L3,L4 and compare with the guided wavelength at four resonance frequencies.

2. Why the coefficient of S2 is 0.2 in line 157?

3. In Fig.3 and Fig. 8, the values of current density should be given.

4. Please denote X_Antenna, X_strip , Y_Antenna and Y_strip in Fig.8. Please explain the equations in Fig 8(a)(b)(c)(d) .

5. Fig. 11(b) should be the transmission coefficient instead of Isolation (dB).

Author Response

The authors would like to thank Editor and all the reviewers for carefully revising the paper and giving constructive suggestions to improve the quality of the paper. Our response to each reviewer was submitted. 

Reviewer 3 Report

There is no further comment.

Author Response

The authors would like to thank the reviewer for carefully revising the paper and giving constructive suggestions to improve the quality of the paper.